# Molecular Insight into Stereoselective ADME Characteristics of C20-24 Epimeric Epoxides of Protopanaxadiol by Docking Analysis

**DOI:** 10.3390/biom10010112

**Published:** 2020-01-09

**Authors:** Wenna Guo, Zhiyong Li, Meng Yuan, Geng Chen, Qiao Li, Hui Xu, Xin Yang

**Affiliations:** 1School of Pharmacy, Key Laboratory of Molecular Pharmacology and Drug Evaluation (Yantai University), Ministry of Education, Collaborative Innovation Center of Advanced Drug Delivery System and Biotech Drugs in Universities of Shandong, Yantai University, Yantai 264000, China; guowenna1@126.com (W.G.); lzhy0818@126.com (Z.L.); sjuiop@163.com (M.Y.); chloeleeq@126.com (Q.L.); 2School of Chemistry and Chemical Engineering, Yantai University, Yantai 264000, China; Cgeng163@163.com

**Keywords:** ocotillol type ginsenoside epimers, stereoselective ADME characteristics, molecular docking analysis, homology modeling, molecular interaction

## Abstract

Chirality is a common phenomenon, and it is meaningful to explore interactions between stereoselective bio-macromolecules and chiral small molecules with preclinical and clinical significance. Protopanaxadiol-type ginsenosides are main effective ingredients in ginseng and are prone to biotransformation into a pair of ocotillol C20-24 epoxide epimers, namely, (20*S*,24*S*)-epoxy-dammarane-3,12,25-triol (24*S*-PDQ) and (20*S*,24*R*)-epoxy dammarane-3,12,25-triol (24*R*-PDQ) that display stereoselective fate in vivo. However, possible molecular mechanisms involved are still unclear. The present study aimed to investigate stereoselective ADME (absorption, distribution, metabolism and excretion) characteristics of PDQ epimers based on molecular docking analysis of their interaction with some vital proteins responsible for drug disposal. Homology modeling was performed to obtain 3D-structure of the human isoenzyme UGT1A8, while calculation of docking score and binding free energy and ligand–protein interaction pattern analysis were achieved by using the Schrödinger package. Stereoselective interaction was found for both UGT1A8 and CYP3A4, demonstrating that 24*S*-PDQ was more susceptible to glucuronidation, whereas 24*R*-PDQ was more prone to oxidation catalyzed by CYP3A4. However, both epimers displayed similarly strong interaction with P-gp, a protein with energy-dependent drug-pump function, suggesting an effect of the dammarane skeleton but not C-24 stereo-configuration. These findings provide an insight into stereo-selectivity of ginsenosides, as well as a support the rational development of ginseng products.

## 1. Introduction

Natural biomolecules with rich diversity in chemical structure are important resources for modern drug discovery [1]. Ginsenoside is a class of triterpenoid saponins abundant in ginseng that has been reputed as the king of medicinal herbs and used as a tonic, prophylactic and restorative agent in traditional medicine for thousands of years [2]. Up to date, over 150 natural ginsenosides have been identified and classified along with different aglycones into two major types of oleanane and dammarane, among which the latter is generally sorted into three subgroups as protopanaxadiol (PPD), protopanaxatriol (PPT) and ocotillol [3,4,5]. The difference in spatial orientation of C-20 hydroxyl further yields a pair of epimers for each, and those in fresh ginseng are usually of the 20*S*-configuration, while processed ginseng products, such as red ginseng, contain both 20*R-* and 20*S*-epimeric forms [6,7].

Pharmacokinetic studies have demonstrated very limited oral bioavailability (<5%) of ginsenosides, for which poor oral absorption and extensive metabolism may be mainly responsible, thus, the aglycone is an actual effective motif through the hydrolysis of sugar moieties by gastric acid or microorganisms in the stomach or enteric canal [8]. Such is the case with the most abundant ginsenoside Rb that can be easily converted into more pharmaceutically potent minor ginsenosides such as Rg3, Rh2 and PPD [9,10]. The aglycone PPD is prone to further oxygenation in vivo to produce intermediate metabolites, a pair of ocotillol-type epimeric C20-24 epoxides (Figure 1), namely (20*S*,24*S*)-epoxy-dammarane-3β, 12β, 25-triol (24*S*-PDQ) and its 24*R*- epimer [4,11,12].

The different stereochemistry of PDQ epimers leads to significant stereoselectivities in both pharmacological effects and ADME properties. More concretely, 24*R*-epimer was found to be equipotent with PPD in terms of attenuating myocardial ischemic injury induced by isoproterenol and hence contribute to overall therapeutic effect of PPD, while 24*S*-PDQ orally administered at an equal dosage had no such effect [13]. In contrast to the 24*R*-epimer, 24*S*-PDQ displayed a higher apparent formation rate from PPD along with a lower elimination rate by phase I metabolism enzymes [14]. Based on assay of in vivo biliary excretion, it was found that 24*S*-PDQ was more preferential to be metabolized into glucuronide conjugates than the 24*R*-epimer, whereas CYP3A4 was confirmed as the predominant isoform responsible for rapid oxygenation metabolism of 24*R*-epimer in human liver microsomes [14,15]. However, to the best of our knowledge, the molecular mechanisms underlying such stereoselective ADME fate of PDQ epimers are still unclear yet.

Our present work aimed to investigate possible mechanisms involved in stereoselective ADME properties of PDQ epimers by molecular docking, which has been demonstrated as an invaluable tool in structural molecular biology and computer-assisted drug design to predict the predominant ligand–protein binding modes for virtual screening on large libraries of compounds and exact three-dimensional (3-D) structures of proteins [16]. The findings would give some insights into stereoselective biological effects of these epimeric intermediate metabolites of naturally abundant ginsenosides and, thus, help to better understand potential mechanisms responsible for clinical efficacy and safety of ginseng products.

## 2. Materials and Methods

### 2.1. Selection of ADME-Related Protein Targets

Three types of enzymes were selected as target proteins for molecular docking analysis of stereoselective ADME properties, which included P-glycoprotein (P-gp, PDB ID: 5KOY), the cytochrome P450 (CYP) isoform 3A4 (PDB ID: 1W0F) and UDP-glucuronosyltransferase (UGT) isoform 1A8 (accession ID: Q9HAW9). According to the available results from both in vitro and in vivo experiments [14,17,18], the three target proteins are mainly involved in disposition of PDQ epimers and other ginsenosides such as absorption, phase I redox metabolism and phase Ⅱ conjugation.

### 2.2. Homology Modeling and Model Validation of UGT1A8

The homology modeling approach in the aid of Maestro (v 10.1, Schrödinger, LLC, New York, NY, 2015) was adopted to predict 3-D structure of the target protein human UGT1A8 for molecular docking analysis since it is not available until now. Firstly, the primary amino acid sequence was retrieved as a target from the NCBI database with an accession number of Q9HAW9 and downloaded as a FAST ALL format file. Then BLAST (the program Basic Local Alignment Search Tool) was performed to find the best homologous sequence with known 3-D structure as template according to the alignment with target by pairwise comparison using BLOSUM62 matrix (Belhesda, LLC, Rockville, MA, USA). Based on the optimal template, homology model was generated with the automated protein structure homology-modeling server Prime Module using a default model. The homology model included copying backbone atom coordinates for aligned regions and side chains of conserved residues, loop modeling and optimization of side chains, building insertions and closing deletion in the alignment. After secondary structure predictions using the Prime module for alignment, the model was subjected to check for any missing side chain and all-atom ab initio energy minimization for refinement, as well as validation according to Ramachandran plot and calculation of RMSD (root-mean-square deviation) by using the Proteins program and Protein Structure Alignment panel, respectively.

### 2.3. Protein Preparation and Optimization

The 3-D structure of UGT1A8 was generated by using Maestro (v 10.1, Schrödinger, LLC, New York, NY, USA, 2015) according to the aligned sequence of the homology model. For the target proteins P-gp and CYP3A4 that have known crystal structures of protein-ligand complexes collected in Protein Data Bank (PDB), the 3-D structures with high resolution and good ligand similarity to the small molecules were selected for further docking. Then optimization of all the three proteins was performed by using Protein Preparation Wizard of Master, including hydrogen bonding network optimization via H-bond assignment section, restrained minimization by adding hydrogen atoms or filling missing side chains, and energy minimization-based refinement via harmonic penalty constraints. After hydrogen addition and removal of water molecules and ligands bound to the complex, the energy-minimized structures of target proteins were finally obtained by using OPLS2005 force field.

### 2.4. Ligands Preparation and Optimization

The PDQ epimers shown in Figure 1 were used as target ligands for docking study. After edited by the 2-D Sketcher option in Maestro (version 10.1) and converted into 3-D structure, the ligands were subjected to refinement in LigPrep module for energy optimization by using the OPLS2005 force field. Then molecular ionization yielded possible states at a target pH value of 7.0 ± 2.0, and the Epik module was performed to generate ligand conformation with the lowest energy, which was used as starting point for further docking experiments.

### 2.5. Binding Site Selection

The structural property, such as active site of target protein, is an important factor for ligand-receptor interaction. Sitemap is a useful module in the Schrödinger suite for binding site identification by characterizing the receptor and ligand regions and hydrophobic interaction forces [19]. According to large-scale verification tests, Sitemap is able to identify binding site bound to ligand molecule with fairly high accuracy (>96%) [20]. Therefore, Sitemap was used to predict optimal binding sites for the proteins UGT1A8 and P-gp, since their active sites ligand for ligand binding are unknown yet. For CYP3A4, the heme pocket has been determined as the active site [21], which thus was selected to investigate ligand binding.

### 2.6. Molecular Docking

All the ligand–protein docking studies were performed with the Glide module within Schrödinger (version 6.7) by using the standard precision (SP) mode that is generally efficient and accurate for most of the targets [22,23]. The molecular mechanics energies combined with the generalized Born and surface area continuum solvation (MM/GBSA) method, one of the popular approaches to estimate free energy of the binding of small ligands to biological macromolecules [24], was used via distance-dependent electrostatic treatment with OPLS2005 force field set to 1000 iterations at the maximum in Prime. The optimal ligand conformations for docking were screened via Emodel module in light of the GlideScore, a mixture of interaction energy and parameter-based penalty functions that roughly represents binding free energy (ΔG_bind_) [25]. The ΔG_bind_ that permits accurate analysis of contribution from each residue by decomposing ligand–protein interaction into different terms [26,27] was calculated to quantitatively assess and rank the ligand–protein binding modes. The docking results were clustered for further analysis, including binding type, the residues involved and some empirical scoring function. Both statistics of GlideScore and ΔG_bind_ were obtained according to triplicate docking analyses with various conformations of the ligand-recepter complexes, and the results were presented as the mean ± standard deviation (SD).

## 3. Results

### 3.1. Homology Modeling and Validation of UGT1A8

Glucuronidation mediated by UGTs is of major importance in the conjugation and subsequent elimination of potentially toxic xenobiotics and endogenous compounds, and more than 35% of phase II drug metabolism is catalyzed by UGT isoforms [28]. To date, nineteen human UGT isoforms have been identified and categorized into three major subfamilies as UGT1A, 2A and 2B according to the sequence homology. It has been demonstrated that the isoenzyme UGT1A8 mainly catalyzes glucuronidation in the intestine and has great potential to influence the pharmacokinetics and biological effects of oral drugs and xenobiotics [29,30,31,32]. The study based on determination of bile excretion in rats further revealed that 24*S*-PDQ was more prone to glucuronidation than its 24*R*-epimer [14]. Therefore, it is worthwhile to investigate ligand–protein binding mode between PDQ epimers and UGT1A8 involved in their stereoselective glucuronidation.

Homology modeling was carried out for further molecular docking analysis since no relevant data of exact 3-D structure of human UGT1A8 has been reported so far. At first, a primary amino acid sequence (accession ID: Q9HAW9) with 399 amino acids was obtained as the target for modeling by BLAST homology searching. Then the crystal structure of macrolide glycosyltransferase was screened from PDB (ID: 2IYA) as the best template for modeling. The protein 2IYA displayed a sequence homology above 37% and the similarity to UGT1A8 could be reach up to 33% (Figure 2), indicating a reliable template for homology modeling in view of the rule of thumb that a sequence homology of 30% or above is adequate [27].

Model validation to check any potential error or deviation from the normal protein is of the utmost importance for further structure-based docking study [33,34]. The alignment score of 0.023 and RMSD value of 0.753 Å (Figure 3a) indicated a good alignment of multiple amino acid sequences between the template protein of 2IYA and homology model of UGT1A8, as well as the reasonability of the obtained homology model. The Ramachandran plot (Figure 3b) generated by Maestro software further shows phi–psi torsion angles for all residues in the structure, among which 96.99% were in the favorable or allowed regions and more than 90% of the backbone dihedral angles resided in the favorable region (in red), whereas the residues in the disallowed region (in white) made up a significantly low share (3.01%).

The 3-D structure of the active site in UGT1A8 was established according to the homology model since it is of the utmost importance for protein-ligand interaction [35]. The active site prediction was performed by using Sitemap. The shade region in Figure 4a shows a 3-D map of the active site within the homologous model protein with the hydrophobic region in red, the acceptor region in blue and the ligand region in yellow, respectively. Figure 4b further illustrates that the ligand–protein interaction occurred in this active region, where all the residues involved resided in the favorable region, while those located in the disallowed regions were far from the active sites and produced little or no effect on UGT1A8 binding with PDQ epimers. All these findings clearly demonstrated structure rationality of the homology model of UGT1A8 with satisfactory geometry and stereo-chemistry for docking studies.

### 3.2. Stereoselective Interaction Between PDQ Epimers and UGT1A8

The homology model of UGT1A8 was used as a receptor for molecular docking analysis of the interaction with PDQ epimers. As shown in Table 1, both the docking score and ΔG_bind_ were negative for each interaction, suggesting stable ligand–protein complexes of both epimers. More concretely, the docking score and ΔG_bind_ were −(4.268 ± 0.174), −(49.89 ± 2.06) kcal/mol for 24*S*-PDQ, and −(3.794 ± 0.208), −(45.11 ± 1.58) kcal/mol for 24*R*-epimer respectively, suggesting significant differences between the two epimers in interaction with the target protein UGT1A8. In contrast to 24*R*-PDQ, the 24*S*-epimer showed a much higher docking score and ΔG_bind_ value, indicating a higher affinity of 24*S*-PDQ with the conjugation isoenzyme of UGT1A8. According to the previously reported in vivo study in rats, the two epimers at the same intravenous dosage indeed displayed stereoselective biliary excretion, and the biliary excretion ratio of 24*S*-PDQ glucuronide was more than 28-fold higher than that of 24*R*-epimer glucuronide [14]. Thus, it is supposed that 24*S*-PDQ, but not 24*R*-epimer, is prone to rapid in vivo glucuronidation, which may be mainly attributed to its high affinity with the UGT isoform of 1A8.

The differentiation between PDQ epimers in their interaction with UGT1A8 was further expounded according to 2-D mappings from molecular docking analysis. Both epimers were anchored to the target protein mainly by hydrogen bonding and hydrophobic interaction, and the electrostatic attraction and van der Waals force could also be observed (Figure 5). Moreover, the stereochemical difference in C-24 indeed caused significantly different interaction patterns, especially the hydrophobic interactions. Both epimers formed three hydrogen bonds with UGT1A8 and showed similar hydrogen bond interactions. For 24*S*-PDQ, the hydroxyl groups at C-3, C-12 and C-25 formed hydrogen bonds via the residues Asp356, Gly274 and Glu377, respectively. Whereas the 24*R*-epimer formed hydrogen bonds via the C-12 and C-25 hydroxyl groups with the residues Glu288 and Gly274, and the third hydrogen bond was observed between the residue Asn276 and the oxygen atom in tetrahydrofuran, but not the C-3 hydroxyl group.

Meanwhile, the PDQ epimers also showed difference in hydrophobic interaction with UGT1A8 (Figure 5a-2, b-2). Although both involved the same residues such as Asn276, Asn355, Asp356 and Asn381, the total number of residues involved was 15 for 24*S*-PDQ, but only 7 for its 24*R-* epimer (including Cys277, Phe287 and Ala289), which contributed to relatively stronger interaction between UGT1A8 and 24*S*-PDQ and was mostly responsible for glucuronidation preferentiality of this 24*S*- epimer. Based on these findings, it thus could be assumed that UGT1A8 may be a major isoenzyme mediating 24*S*-PDQ conjugation with glucuronic acid and the C-24 stereochemistry plays a great role in distinguishing glucuronidation metabolism of PDQ epimers.

### 3.3. Stereoselective Interaction between PDQ Epimers and CYP3A4

The cytochrome P450 superfamily (CYP) is a group of mono-oxygenases participating in the metabolism of both endogenous and exogenous substances, among which CYP3A is the most abundant subtype in human liver. By using chemical inhibition in human liver microsomes and recombinant human P450 isotype determination, previous studies have manifested that CYP3A4 is the major isoform responsible for difference in oxidative metabolism of PDQ epimers [12]. Herein, human CYP3A4 was used as a target protein to investigate stereoselective redox metabolism of PDQ epimers.

CYP3A4 is the most abundant xenobiotic-metabolizing cytochrome P450 isoform that contains a heme co-factor as the active site. A number of researches have clearly demonstrated the broad substrate specificity of the CYP3A4 molecule, as well as the promiscuity of ligand binding in the CYP3A4 heme pocket [21]. CYP3A4 is known to bind different sizes of compounds such as metyrapone (226Da), progesterone (314Da), bromocriptine (655Da) and cyclosporine (1203Da). Intriguingly, metyrapone bound via a pyridine nitrogen to heme iron, while progesterone did not locate in the active heme pocket but was observed on a peripheral site of the CYP3A4 molecule [36]. Taking into consideration that the molecular structure of the tetracyclic triterpenoid PDQ epimers is similar to the natural steroid hormone progesterone, the X-ray crystal structure containing the progesterone with the PDB ID of 1W0F thus was chosen for molecular docking analysis in the present study.

As shown in Table 1, the docking score and ΔG_bind_ value were −(5.737 ± 1.35) and −(54.95 ± 5.27) kcal/mol for 24*S*-PDQ, and −(7.162 ± 0.040) and −(61.65 ± 0.57) kcal/mol for 24*R*-epimer, respectively. Such negative and high values also demonstrated that both epimers could spontaneously form thermodynamically stable complexes with CYP3A4. The difference further indicated considerable differentiation in interaction with the protein CYP3A4 that was closely related to the different C-24 stereochemistry. In contrast to the 24*S*- epimer, 24*R*-PDQ showed higher docking score and ΔG_bind_ value, suggesting a stronger affinity to CYP3A4.

The 2-D plots clearly illustrate the difference in the interaction mode (Figure 6). Intriguingly, the heme co-factor of CYP3A4 only formed Fe-S coordination with the residue Cys442, but no coordination was observed with the ligand molecule of either 24*S*- or 24*R*-PDQ, which was highly consistent with the ligand binding modes of CYP3A4 previously reported [21]. More concretely, both epimers were found on a peripheral site of the CYP3A4 molecule mainly via a hydrogen bond and hydrophobic interaction. Most of the amino acid residues involved in hydrophobic interaction were identical for the two epimers, such as Phe57, Arg105, Arg106, Ser119, Ile120, Arg212, Phe215, Phe304, Ala305, Thr309, Ile369, Ala370, Arg372, Leu373 and Glu374. However, a significant difference between two epimers could be found in hydrogen bonding mode. Although 24*S*-PDQ did not form any hydrogen bond with CYP3A4, the 24*R*- epimer formed a hydrogen bond via C3-OH with the residue Glu374, which thus caused relatively stronger interaction between CYP3A4 and 24*R*-PDQ and was responsible for the differences in docking score and ΔG_bind_ value. This finding provided a new evidence for promiscuity of ligand binding in the CYP3A4 heme pocket [21,36], and also demonstrated the great effect of C-24 stereochemistry on how PDQ epimers interact with CYP3A4, also proved that 24*R*-PDQ was more prone to in vivo oxidation catalyzed by CYP3A4 [12].

### 3.4. Stereoselective Interaction between PDQ Epimers and P-gp

P-gp is a well-known protein with energy-dependent drug-pump function that shows significant effect on drug fate in vivo [37]. The research work of Zhang et al. revealed the differential regulations of P-gp by ginsenoside Rh2 epimers in vivo, which provided new evidence of the chiral characteristics of this protein and was helpful to elucidate the stereoselective P-gp regulation mechanism of ginsenoside epimers from a pharmacokinetic view [17]. Interestingly, it was further reported that both 24*S*- and 24*R*-PDQ, which share the same dammarane skeleton, could be distinguished by P-gp and had similar inhibitory effects on P-gp by decreasing efflux of digoxin across Caco-2 cell monolayers [38]. P-gp thus was selected as a target protein in our present study for ligand–protein interaction analysis to investigate possible molecular mechanism responsible for such inhibitory effect of PDQ epimers on this stereoselective biomacromolecule P-gp, and the X-ray crystal structure with an ID of 5KOY in PDB was selected for molecular docking.

The molecular docking analysis revealed docking score and ΔG_bind_ value of −(5.636 ± 0.033) and −(42.73 ± 0.58) kcal/mol for 24*S*-PDQ, and −(5.879 ± 0.380) and −(40.73 ± 0.77) kcal/mol for 24*R*-PDQ, respectively (Table 1). Such high and similar values demonstrated that both epimers could spontaneously form stable complexes with P-gp, and these octillol type molecules may both be potent P-gp inhibitors with similarly high affinity to P-gp, which was not affected by the difference in C-24 stereochemistry.

Further interactions mode analysis by using 2-D mappings (Figure 7) clearly demonstrated that PDQ epimers could form stable complexes with P-gp primarily through hydrophobic interaction, but not any hydrogen bond, which was very different from their interactions with the protein molecules UGT1A8 or CYP3A4. More concretely, the hydrophobic binding pockets in P-gp were almost the same for embedding 24*S*- and 24*R*-PDQ, which mainly consisted of residues such as Leu857, Ile860, Val861, Ile864, Ala865, Gly868, Met945, Ser948, Tyr949, Cys952, Phe974, Val978, Ala981, Met982, Gly985 and Ser989. This result provided an additional evidence for similar P-gp inhibition potency of PQD epimers from the viewpoint of similar hydrophobic interaction with the P-gp molecule. The previously reported studies based on Western blot analyses demonstrated such a structure–activity relationship, i.e., that PDQ epimers and 20*S*-Rh2 shared the same dammarane skeleton as the pharmacophore responsible for P-gp inhibiting activity, while the ocotillol side chain and C-24 stereo-configuration of PDQ epimers had no effect on the expression of P-gp in Caco-2 cells [38]. The findings from in silico molecular docking analysis were highly consistent with those from in vitro cell-based assays and brought a further insight into molecular mechanism of P-gp inhibiting activity of gesenosides.

## 4. Conclusions

Chirality is a common phenomenon and exploring interaction between biomacromolecules and chiral small molecules has preclinical and clinical significance. Natural protopanaxadiol-type ginsenosides are usually present in 20*S*-configuration and prone to oxidation into a pair of chiral ocotillol-type C20-24 epoxides, 24*S*-/24*R*-PDQ epimers. Several previous studies have revealed their stereoselective fates in vivo. Our present work brought some new insight into the molecular mechanism involved in stereoselective ADME characteristics of PDQ epimers by performing molecular docking analysis of their stereoselective interaction with those vital bio-macromolecules involved in drug disposal such as UGT1A8, CYP3A4 and P-gp. The findings were in good agreement with those cell/animal-based experimental observations and provided new evidence for stereoselectivity of structurally diverse gesenosides from the viewpoint of ligand–protein interactions.

## Figures and Tables

**Figure 1 biomolecules-10-00112-f001:**
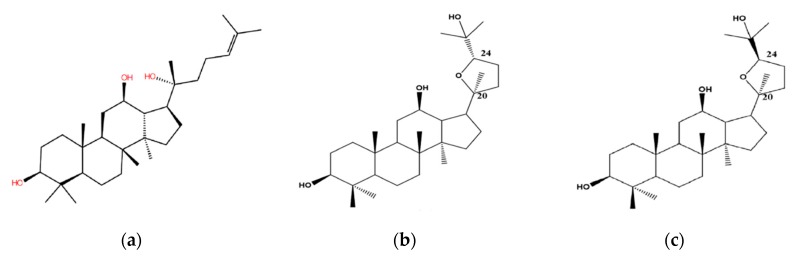
Chemical structure of (**a**) 20*S*-PPD; (**b**) 24*S*-PDQ; (**c**) 24*R*-PDQ.

**Figure 2 biomolecules-10-00112-f002:**
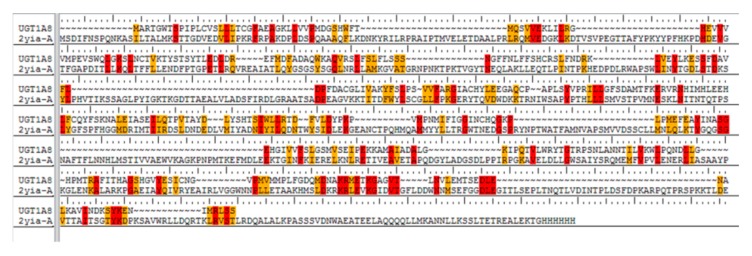
Sequence alignment between UGT1A8 query sequence and the template amino acid sequence of 2IYA. All the identical residues were highlighted in red color, and those similar residues in orange color respectively.

**Figure 3 biomolecules-10-00112-f003:**
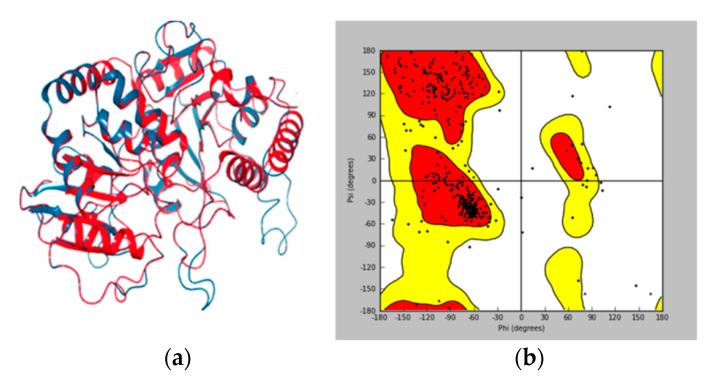
Homology model validation. (**a**) RMSD between the template (in blue) and target protein (in red). (**b**) Ramachandran (phi/psi) plot. Red: favorable region; yellow: allowed region; white: disallowed region.

**Figure 4 biomolecules-10-00112-f004:**
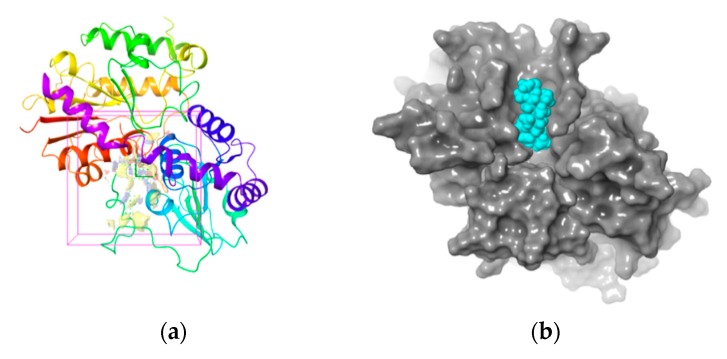
(**a**) Three-dimensional structure of active site within the homology model of UGT1A8 (region in the shadow). Stripe structure: the homology model of UGT1A8; Red: the hydrophobic region; blue: the acceptor region; yellow: the ligand region. (**b**) 3-D map of interaction between PDQ and UGT1A8. Gray region: UGT1A8; blue region: PDQ.

**Figure 5 biomolecules-10-00112-f005:**
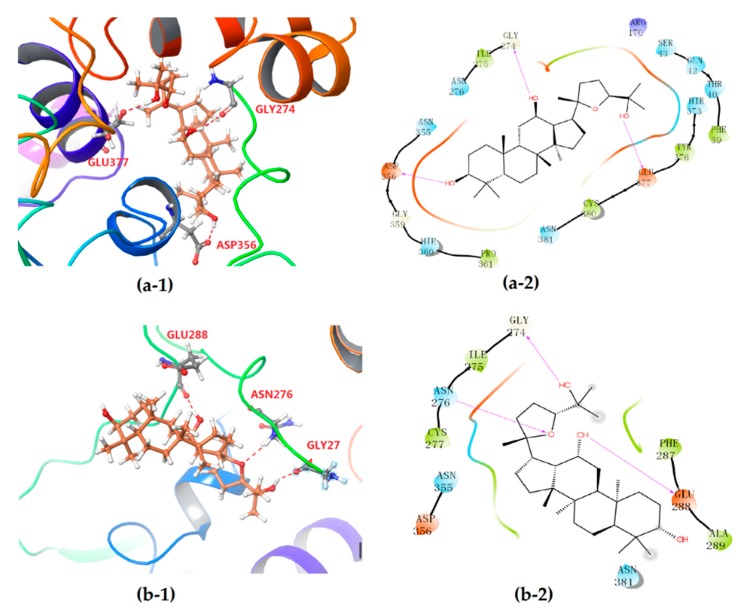
3-D (**1**) and 2-D (**2**) diagrams illustrating the molecular interaction of 24*S*-PDQ (**a**) and 24*R*-epimer (**b**) with the modeled UGT1A8. (**1**) The ball-stick structure: PDQ; the cartoon stucture: UGT1A8. (**2**) Purple arrow: hydrogen bond; red: charged (negative); purple: charged (positive); white: glycine; green: hydrophobic; blue: polar.

**Figure 6 biomolecules-10-00112-f006:**
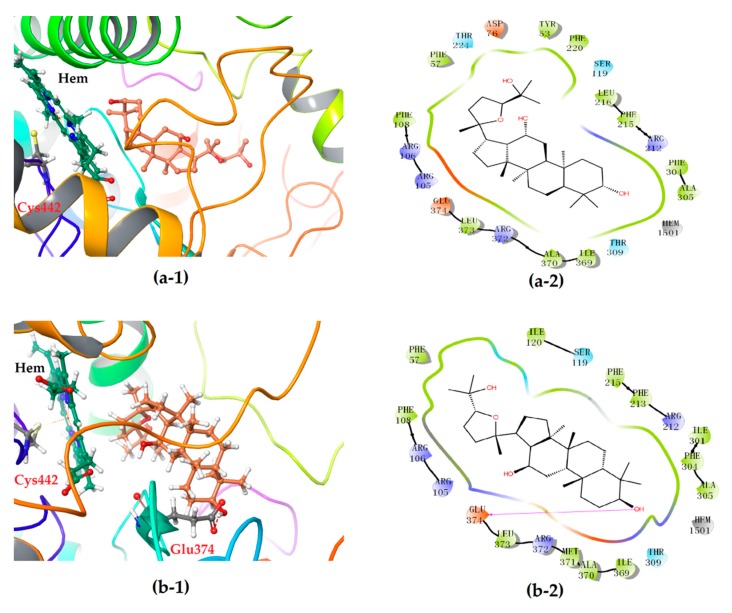
3-D (**1**) and 2-D (**2**) diagrams illustrating the molecular interaction of 24*S*-PDQ (**a**) and 24*R*-epimer (**b**) with the CYP3A4. (**1**) The ball-stick structure: PDQ; the cartoon stucture: CYP3A4. (**2**) Purple arrow: hydrogen bond; red: charged (negative); purple: charged (positive); white: glycine; green: hydrophobic; blue: polar.

**Figure 7 biomolecules-10-00112-f007:**
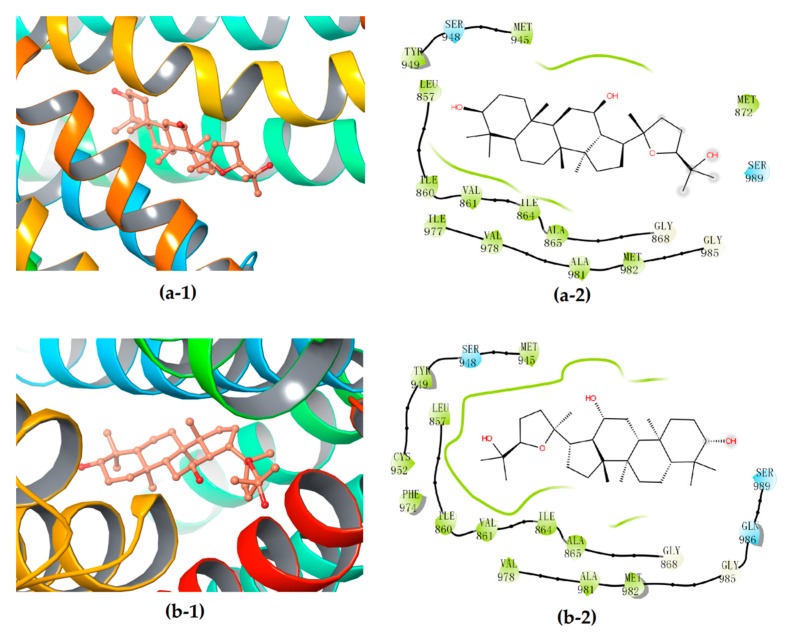
3D (**1**) and 2D (**2**) diagrams illustrating the molecular interaction of 24*S*-PDQ (**a**) and 24*R*-epimer (**b**) with the P-gp. (**1**) The ball-stick structure: verapamil; the cartoon stuctures: P-gp. (**2**) White: glycine; green: hydrophobic; blue: polar.

**Table 1 biomolecules-10-00112-t001:** Molecular docking statistics for interaction between PDQ epimers and various proteins ^1^.

Protein-Ligand Interaction		Docking Score	ΔG_bind_ ^2^	Residues for Hydrogen Bonding
**UGT1A8**	24*S*-PDQ	−(4.268 ± 0.174)	−(49.89 ± 2.06)	Gly274, Asp356, Glu377
	24*R*-epimer	−(3.794 ± 0.208)	−(45.11 ± 1.58)	Gly274, Asn276, Glu288
**CYP3A4**	24*S*-PDQ	−(5.737 ± 1.350)	−(54.95 ± 5.27)	/
	24*R*-epimer	−(7.162 ± 0.0395)	−(61.65 ± 0.572)	Glu374
**P-gp**	24*S*-PDQ	−(5.636 ± 0.0326)	−(42.73 ± 0.577)	/
	24*R*-epimer	−(5.879 ± 0.380)	−(40.73 ± 0.774)	/

^1.^ Glide module within Schrödinger (version 6.7, LLC, New York, NY, USA, 2015) was performed, and the data of docking score and binding free energy were presented as mean ± SD of triplicate docking analyses with various conformations. ^2.^ ΔG_bind_ represents the binding free energy in kcal/mol calculated using the Prime/MM-GBSA protocol.

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
