# Peer review of "Molecular Insight into Stereoselective ADME Characteristics of C20-24 Epimeric Epoxides of Protopanaxadiol by Docking Analysis"

_biomolecules, 2020, doi:10.3390/biom10010112_

Round 1

Reviewer 1 Report

found the idea of the manuscript interesting and promising. However, there are some major issues which I think need to be addressed. I think, some parts of the work needs to be redone.

I have prepared a lot of notes of small corrections of the text until I became aware of the larger problems, so I will focus on those only.

The main question to ask: do I, or the reader believe in the results of docking? For about half of the work I am afraid the answer is ‘no’. And it should be the authors who should question everything!

Let’s look at the first receptor, UGT1A8. The PDQ ligand is not too flexible, which is good (easier to dock since fewer errors creep in). The docking pocket seems to reasonable, and, importantly, the structures of the two epimers has many similarities (the binding modes partially overlap, which is good since the large part of both epimers is identical), which is also good and tells that we could be on the right track. So I think this is a reasonable part of the work.

My only remark here is that the interaction plots (For example Figure 5  b-2 and a-2) are very difficult to compare. It would be nice to position them in such way the differences can be immediately seen.

The second receptor, CYP3A4, is a problem. The protein contains heme co-factor, and the ligands in the PDB as far I have seen are all coordinated with iron atom. (If the authors can show structures which this does not happen, I will happily withdraw this criticism). In your docking the heme is not even in the binding site. You should note that Glide docking does  not necessarily correctly ligates ligand to the metal. What you need is “covalent docking”. It’s possible to do something like that in Glide, but it’s not automatic (you have to specify ligating atoms manually, as far as I know); in addition, the problem is that it is probably not known which ligand atom is ligated to the heme. So this is a tough problem! Unfortunately, I don’t believe the authors were able to solve it in this manuscript.

The third receptor P-gp is also a problem, but in a different way. Here, we have a relatively rigid ligand (PDQ derivative), which is good, see above. The second ligand, verapamil, is very long and flexible, and this is a usually a problem: in my experience, long and flexible ligands were very hard to dock correctly. To make matters much worse, the binding site is quite large. Because of the latter, it is nearly impossible to dock correctly verapamil. A sign that the docking was probably incorrect is that the binding modes are very different, while some sort of indication to a converged  docking solution would be a partial overlap of the solutions.

Even for PDQ derivative, the ligand could go to a lot of places, and I wouldn’t bet that the shown binding mode is correct. The docked energy of the best hit could be accidental. I wonder how different are the binding energies of hits number 2,3,4, etc. If they are relatively close to the binding affinity to hit number 1, this would be an indication that the ligand does not care where to go in the binding site due to the size of the binding pocket, and what I was saying about doubtful best conformation is correct. Perhaps if you could see clearly then that for example R-epimer hits  number 1,2,3,4, etc are consistently binding better than S-epimer, this could be an indication that R-epimer indeed binds stronger than S-epimer, even if the binding mode is varying.  Or perhaps trying to compare R- and S- ligand binding modes side by side, narrowing down to the similar binding modes, and comparing how the energies between R- and S- ligands differ, might lead to useful insights.

Therefore, I recommend to think more about your docking results, especially for the P-gp, and create a better docking set up for the CYP3A4 receptor, which would require quite a bit of rework.

I do understand that the authors might think: everybody docks just like we did. The truth is, it is system dependent. Some of the systems the authors chose are very challenging (nobody said it's easy), and the authors should tread very carefully.

Reviewer 2 Report

In this manuscript, the authors study the stereoselectivity of two gingenoside epimers.

The rational of their study is well described in the Introduction part and appears as relevant.

Based on differences found between the biological activities of the two epimers, the authors study their molecular interactions with three enzymes involved in the absorption and metabolic processes. The study is based on former results obtained on differences in the biological or pharmacological different properties of the two epimers. In this manuscript, the authors analyse the molecular docking of the compounds with models of the enzymes or homology models for UGT1A8.

The analyses appear to be sound and the authors correlate the different results which are obtained using such docking approaches with stereoselectivity already observed in pharmacological models. As such, the study presented in the manuscript is a complement of former results already reported and well referenced in the appropriate section.

Comments:

The need of the authors to report results obtained with verapamil on P-gp is not clear (Figure 7 and lines 276 to 289) and might be omitted if no further explanations are given. Legend of Figure 8 has to be corrected since verapamil does not appear in the molecular docking.

Round 2

Reviewer 1 Report

The authors made a significant effort to address the issues raised in the first stage of the review process, which I commend.

I have only short question (correction?):

Line 133: “were screened via Emodel module in light of the GlideScore,”: what does “in light” mean? Based on this sentence, I wouldn’t be able if Emodel is from Schrodinger of not.